# Connection of ssDNA to Silicon Substrate Based on a Mechano–Chemical Method

**DOI:** 10.3390/mi14061134

**Published:** 2023-05-28

**Authors:** Liqiu Shi, Feng Yu, Mingming Ding, Zhouming Hang, Yan Feng, Aifang Yan, Hongji Dong

**Affiliations:** 1School of Mechanical and Automotive Engineering, Zhejiang University of Water Resources and Electric Power, Hangzhou 310018, China; 2Key Laboratory for Technology in Rural Water Management of Zhejiang Province, Zhejiang Engineering Research Center for Advanced Hydraulic Equipment, Hangzhou 310018, China

**Keywords:** mechano–chemical method, single crystal silicon, diazo salt of benzoic acid, coupling layer, ssDNA

## Abstract

A novel fabrication process to connect single-stranded DNA (ssDNA)to a silicon substrate based on a mechano–chemical method is proposed. In this method, the single crystal silicon substrate was mechanically scribed in a diazonium solution of benzoic acid using a diamond tip which formed silicon free radicals. These combined covalently with organic molecules of diazonium benzoic acid contained in the solution to form self-assembled films (SAMs). The SAMs were characterized and analyzed by AFM, X-ray photoelectron spectroscopy and infrared spectroscopy. The results showed that the self-assembled films were covalently connected to the silicon substrate by Si–C. In this way, a nano-level benzoic acid coupling layer was self-assembled on the scribed area of the silicon substrate. The ssDNA was further covalently connected to the silicon surface by the coupling layer. Fluorescence microscopy showed that ssDNA had been connected, and the influence of ssDNA concentration on the fixation effect was studied. The fluorescence brightness gradually increased with the gradual increase in ssDNA concentration from 5 μmol/L to 15 μmol/L, indicating that the fixed amount of ssDNA increased. However, when the concentration of ssDNA increased from 15 μmol/L to 20 μmol/L, the detected fluorescence brightness decreased, indicating that the hybridization amount decreased. The reason may be related to the spatial arrangement of DNA and the electrostatic repulsion between DNA molecules. It was also found that ssDNA junctions on the silicon surface were not very uniform, which was related to many factors, such as the inhomogeneity of the self-assembled coupling layer, the multi-step experimental operation and the pH value of the fixation solution.

## 1. Introduction

In the construction of DNA biosensors and the manufacture of DNA chips, the effective fixation of DNA probe on the surface of the converter or carrier is an important basic premise. Therefore, the research on DNA fixation has important guiding significance for the improvement of sensor and chip technology [1,2,3]. Typical methods of DNA probe fixation include adsorption, SAM (self-assembling film), and covalent bonding. Among them, the SAM method and covalent bonding method can produce a stable modification layer and improve the firmness and durability of the probe [4,5,6,7]. Fang et al. [8] self-assembled amino thiols on the surface of a gold electrode to introduce amino groups. In the presence of a chemical coupling activator, water-soluble carbodiimide (EDC), the probe DNA molecules were covalently fixed on the electrode surface through a condensation reaction. Mirsky et al. [9] used a gold electrode modified with alkyl thiols to covalently fix NH_2_–DNA and studied the influence of different fixation conditions on the fixation density. The self-grouping covalent fixation method of the gold surface does not require special surface treatment, so it is a relatively simple method. However, the limitation is that it can only be applied to the gold surface.

Among the substrates used for DNA chips, the most common are glass slides and silicon substrates. Compared with glass, silicon substrate has high thermal conductivity, good finish and can withstand high temperatures. Silicon has been extensively explored to address multidrug resistant bacterial diseases and to enable developments in novel and innovative approaches for next-generation highly efficient, cost-effective, and reliable multifunctional biomedical tools [10]. Wang et al. [11] produced modulated light-activated electrochemistry on silicon for addressable biosensing. To stabilize and functionalize the silicon substrate, metal–organic framework (MOF) nanoparticles were grown in-situ on the silicon electrode. The photocurrent increased due to the enhanced reduction process after DNA binding. This work provided a promising platform for multi-spot and label-free DNA chips. Blaschke et al. [12] prepared silicon substrates with stripe-patterned surface-near electrostatic forces (SNEF) by local implantation of boron ions into n-type silicon wafers and phosphorus ions into p-type silicon wafers in a stripe pattern of 12 μm periodicity. The negatively charged single stranded deoxyribonucleic acid (ssDNA) and bovine serum albumin (BSA) proteins were immobilized on silicon regions. Quake et al. [13] prepared single-molecule DNA chips by laying a dilute solution onto silicon wafers. The randomness of single-molecule DNA chips prepared by this method leads to two problems: one is that some DNA molecules are too dense and exceed the resolution of the instrument; the other is that the molecules in some areas are too sparse, resulting in the waste of limited space. DNA nanosphericity has been reported by Complete Genomics [14]. The substrate was prepared by photolithographic technology. The HMDS layer was prepared on the surface of the silicon substrate, on which the amino-silane nanodot array was regularly distributed for fixing DNBs. Then the DNBs were loaded onto the array to obtain the DNBs chip. Yang et al. [15] tried to fix DNA with biotin at one end and a Cy3 fluorescence group at the other end, onto silicon substrate based on the biotin–streptavidin system to conduct fluorescence observations and verify the feasibility of the method for connecting DNA to a silicon base. Ryu et al. [16] proposed a new biosensor for label-free DNA detection to enhance the sensitivity. A gold nanoparticle(GN) embedded silicon nanowire (SiNW) configuration was prepared. Due to its simple preparation process, high thermal stability, the high immobilization efficiency of the mercapto group in the self-assembled monolayer (SAM), and improved sensitivity, this novel spherical GN and SiNW nanostructure-combined structure has high potential as a label free biosensor.

In summary, at present, the preparation of silicon-based DNA probes is mostly at a two-dimensional level [17,18]. However, this kind of surface has the disadvantage of not being able to fix biomolecules at high density, and still faces great challenges in terms of sensitivity and repeatability, which also restricts the wide application of these DNA chips [19,20]. Compared with a two-dimensional surface, a substrate with a three-dimensional structure can fix more biomolecules in the direction of the vertical surface, which is the basis for realizing the high-density immobilization of biomolecules on the substrate surface.

Therefore, this paper attempts to propose a new technique to link ssDNA on a silicon substrate with aromatic diazo salt molecules as a medium. The technology uses a mechano–chemical method to process the three-dimensional controllable structure on the silicon surface, and self-assemble the aromatic hydrocarbon coupling layer at the same time. Single-stranded DNA can be connected to the 3D structure on the silicon surface through the coupling layer in a covalent binding mode, and finally realize the connection of the ssDNA probe. While fixing ssDNA, the size, position and shape can be highly controlled. This technology has important scientific value, theoretical significance and engineering application prospects for promoting the development of nano-devices, biosensors, disease diagnosis, environmental monitoring and other fields. Our team is trying to use this technology to detect microorganisms corroding flood gates. It will have a wider application space in the future.

## 2. Experimental Process and Methods

### 2.1. Principle of ssDNA Connectionby a Self-Assembled Coupling Layer

DNA can be immobilized on the surface of the support by covalent bonds such as amide bonds, ester bonds and ether bonds [21,22]. The fixation method of amino-modified DNA is a popular method because of its simple operation and short time requirements [23]. In this experiment, the silicon surface was treated in a boron tetrafluoride benzoate diazo salt solution by a mechano–chemical method, and a self-assembled film ending with a carboxyl group (–COOH) was produced on it. The end of the ssDNA was then modified with an amino group (–NH_2_). Finally, under the action of a covalent coupling activator, the amino group of the probe reacted with the carboxyl group on the silicon surface to form an amide bond. The DNA probes were anchored to the silicon surface. The principle is shown in Figure 1.

### 2.2. Controllable Self-Assembly of an Aromatic Hydrocarbon Coupling Layer on a Silicon Surface

The substrate used in this experiment was P-type Si(100) with a thickness of 460 ± 15 µm. In addition, the experiment also used 99.999% pure nitrogen, the resistance of 18.2 MΩ ultra-pure water (Milli-Q water), and distilled water. The specific experimental steps were as follows:(1)The pretreated silicon substrate was fixed in a tank containing 50 mmol/L boron tetrafluoride benzoate diazo salt solution, which was fixed in the constructed mechano–chemical micromachining system.(2)The diamond tool was moved at a certain speed (500 nm/s) and direction by the program of the micro-movement table, so that the silicon surface could be scribed and the marking area could be functionalized at the same time.(3)After scribing, in order to ensure sufficient reaction time for the silicon sample and assembly solution, the silicon wafer was placed in the assembly solution, and kept away from light. After about 12 h, the sample was removed, and rinsed with nitrile, acetone, absolute ethanol, and a large amount of ultra-pure water for detection and characterization.

### 2.3. Connection of DNA via the Coupling Layer

The silicon was scribed in the diazo benzoate solution to generate a self-assembled film. At this time, the end group of the molecular membrane was the carboxyl group (–COOH).Then, under the covalent coupling of N-ethyl-N′-(3-dimethylaminopropyl) carbodiimide hydrochloride (EDC), it reacted with the modified amino group (–NH_2_) at the 3′ end of ssDNA to form an amide bond, which realized the attachment of ssDNA to silicon via the self-assembled coupling film.

The ssDNA used in this experiment was synthesized by Shanghai Sangon Bioengineering Technology Service Co., Ltd. (Shanghai, China). FAM fluorescence was attached at the 5′ end, NH_2_ was modified at the 3′ end, the probe length was 24 bp, and the base sequence from the 5′ end to the 3′ end was GCA AAG GGT CGT ACA CAT CATCAT. The molecular weight was 8063.5 and the net content was 30.0 OD.EDC reagents were purchased from Beijing Bailing way Chemical Technology Co., Ltd. (Beijing, China), and frozen at −20 °C. It should be noted that FAM fluorescence labeled ssDNA in order to detect the attached ssDNA, so all operations using ssDNA were carried out in a dark room. The sample was tightly wrapped with tin foil to avoid fluorescence quenching.

At the same time, in order to verify whether the parts of the silicon surface that have not been functionalized by aromatic hydrocarbons were connected with single-stranded DNA, the silicon surface was washed with 0.2 mol/L NaOH solution and 0.1 mol/LNaCl solution for 30 min, respectively, after reaction.

## 3. Results and Discussion

### 3.1. AFM Characterization

The AFM Dimension 3100 (Digital Instruments, Tonawanda, NY, USA) was used for topography measurement. The surface morphology images of the samples before and after assembly were recorded in contact mode [24,25]. In order to facilitate comparison, all morphology images were scanned with the same tip (V-shaped Si3N4 micro cantilever, length 200 μm, elastic constant 0.12 N/m). The imaging was performed in air at 300 K and relative humidity of 40%. The range of scanning was 3 μm, and the rate of scanning was 1.5 Hz.

The roughness analysis and comparison of silicon substrate before and after assembly in benzoate diazo salt solution is shown in Figure 2. It can be seen that the fine step surface can be observed before assembly, Ra is 6.511 nm. After assembly, the step surface disappeared and a cluster shape appeared. The relatively uniform structure with a Ra of 3.728 nm was formed.

After the characterization by atomic force microscopy, it can be seen that there were differences in morphology and roughness of the scribed area before and after assembly, which proved the existence of the self-assembled film. However, it could not infer whether the assembly molecules were bound to the silicon atoms by chemical, covalent or physical adsorption. Therefore, it was necessary to combine the spectral analysis methods to interpret the binding mode between the organic molecules and silicon atoms using the changes of chemical composition and chemical bonds before and after assembly in the scribed region.

### 3.2. XPS Analysis

X-ray photoelectron spectroscopy was used to analyze the elemental composition, chemical valence state, energy range of physical effects and electronic structure of the sample surface [26]. The PHI 5700 ESCA System produced by the American Physical Electronics Company was used in this detection, and the excitation source was Al Ka (=1486.60 eV).

Figure 3 shows the XPS spectra of the sample before and after functionalizing DNA molecules to the Si substrates. The percentage content of elements on the silicon surface before and after assembly are shown in Table 1. It can be seen from Figure 3 and Table 1 that obvious N1s and P2p peaks appear after DNA assembly. The peak around 398.4 eV is the characteristic peak of N1s which were introduced due to the DNA modified with –NH_2_. The P2p peak around 130.6 eV was caused by the single stranded DNA base sequence. The content of C1s was significantly increased after assembly, mainly due to the benzene rings contained in organic molecules of self-assembled films, which was consistent with theoretical inference. In terms of oxygen content, the original Si–O connection was replaced by the SAMs in the marked area, so the oxygen content showed a tendency to decrease. The content of Si2p was obviously weakened, because some of the silicon atoms had been covered by the SAMs, which led to a decrease in the detected Si content. In general, the content of oxygen and Si2p before and after assembly were consistent with the experimental results. The content changes of main elements suggested that DNA had been attached to the silicon surface via the aromatic hydrocarbon coupling layer.

The fine scanning analysis and differentiation of the Si2p peak is shown in Figure 4. The Si2p peak was divided into two peaks. According to the peak attribution table [27], it can be seen that the Si at 100.38 eV is in the Si–C bond region and the Si at 103.45 eV is in the Si–O bond region [28,29]. The ratio of silicon peak area at 100.38 eV and 103.45 eV before the reaction was about 4:1, which indicated that the silicon surface was dominated by silicon oxide. However, the ratio of the two silicon peak areas after the reaction was about 1:3, which showed that the silicon combined with carbon on the silicon surface was greatly increased. This was due to the combination of diazo organic molecules with the Si substrate by a Si–C bond, and the Si–O bond being replaced by the Si–C bond.

Taking the C1s peak before self-assembly as the benchmark, it can be seen from Figure 5 and Table 1 that the C1s peak is significantly enhanced after assembly, and its peak position remains unchanged. This is mainly due to the benzene ring in the organic molecules formed in the film. The preliminary results show that the benzene ring molecules on the diazo salt have been assembled on the silicon substrate, which is consistent with the theoretical inference.

In addition, peak fitting of assembled C1s peaks was also carried out, as shown in Figure 5. According to the standard peak value of carbon provided in the literature, the peak around 285 eV is caused by the contaminating carbons inside the silicon wafer and the adsorption of atmospheric carbon elements, the peak around 287.8 eV is the characteristic peak of the carbon atom in C=O, and the peak around 288.9 eV is the characteristic peak of the carbon atom in –COOH. This confirms that the benzoate diazo salt molecules ending in–COOH had been bound to the silicon surface.

### 3.3. FT–IR Analysis

Infrared spectroscopy (IR) is the most important method for structural identification of organic compounds and can effectively help identify functional groups [30]. The Fourier transform infrared spectrometer (FT–IR) model, AVATAR 360, produced by the Nicolet Company of the United States was used in this experiment. The scanning times were 64, and the resolution was 4 wave numbers.

Figure 6 shows the infrared spectra obtained from the blank silicon wafer and the silicon surface modified with diazo benzoate film by KBr Pellets. The blank silicon terminated with oxygen basically had no other obvious absorption peak except for the characteristic strong Si–O–Si absorption band at 1086.54 cm^−1^. This was mainly used for comparative analysis. For the self-assembled film scribed in benzoate diazo salt solution, the structure of benzoate was attached to the silicon surface. According to the position and attribution table of the characteristic absorption peak of aromatic hydrocarbons [31], the peak at 1680 cm^−1^~1620 cm^−1^ was the vibration absorption peak of the connection between an aryl group and –COOH. It was present in the spectrogram.

The electron absorption effect of –COOH caused the absorption wave number of *ν* (C=O) to increase and shift to the area around 3500 cm^−1^. The absorption peak of the O–H stretching vibration was typical, in the region of 3000~2500 cm^−1^, while the blunt and smooth absorption peak near 1100 cm^−1^ was the vibration absorption peak of silicon bonding with an aryl group. It is proven that the diazo benzoate SAMs had been covalently connected to the silicon surface by Si–C. The large difference in peak position, intensity and shape in the 1000~400 cm^−1^ region in Figure 5 was most likely due to the strong electron effect of –COOH, which made the band in the 900~650 cm^−1^ region lose the characteristics of a substituted benzene. This was also an important basis for distinguishing the components of the diazo salt assembly membrane. Infrared spectroscopy not only proved the existence of aromatic hydrocarbon diazo salt assembly molecules on the silicon surface, but also proved that these SAMs were covalently connected to the silicon surface.

### 3.4. Fluorescence Microscope Analysis

The Olympus BX51 positive fluorescence microscope was used to detect the fixed DNA in the self-assembled region. The optimal emission wavelength range of the microscope was above 500 nm, while the maximum emission wavelength (Mission) of FAM fluorescence labeling was 520 nm, which was in line with the detection range [32,33].

As shown in Figure 7, regular straight lines with green fluorescence could be seen under the fluorescence microscope at 400 times magnification, which were all formed by diamond cutting tools. Figure 7a is the AFM image measured after scribing the silicon surface in solution, and Figure 7b is the corresponding fluorescence image after connecting ssDNA. The line-widths of the two lines were 4 μm and 1 μm, respectively. Although the line-widths of the two lines were quite different, the fluorescence signals were relatively strong. This indicated that ssDNA was successfully connected to the self-assembled film on the silicon surface.

In order to prove that only functional areas on the silicon surface were connected to ssDNA, the silicon surface after reaction was cleaned and the results are shown in Figure 7c. Figure 7c is the fluorescence detection image of DNA attached to a three-dimensional cross structure after cleaning. The area with fluorescence signal in Figure 7c was the part scribed in the solution, and the black areas on both sides were the unpainted parts. This method of cleaning and validation of functional groups has been used extensively in the literature [15,19], which indicated that cleaning was very important to exclude false signals. This was a good comparison to show that we can fix DNA by this mechano–chemical method. This will lay the foundation for producing subsequent DNA chips and DNA sensors.

We also studied the influence of ssDNA probe concentration on fixation effect. Figure 8 showed the fluorescence detection results of probe fixation with different concentrations of ssDNA. The magnification of the microscope was 1000 times. As can be seen from Figure 8, with the gradual increase in ssDNA concentration from 5 μmol/L to 15 μmol/L, the fluorescence brightness gradually increased, indicating that the fixed amount of ssDNA increased. However, when the concentration of ssDNA increased from 15 μmol/L to 20 μmol/L, the detected fluorescence brightness decreased, indicating that the hybridization amount decreased. The reason may be related to the spatial arrangement of DNA and the electrostatic repulsion between DNA molecules. This research result was in line with some domestic scholars. For example, Liu et al. [34] showed that the fixed amount of sulfo-modified DNA probe on a gold surface increased with an increase in DNA concentration, the hybridization amount also increased, and the hybridization time was shortened. However, when the DNA probe reached a certain concentration, the hybridization amount decreased. The study of Yamaguchi et al. [35] also showed that the fixed amount increased with an increase in probe concentration, but saturation adsorption would occur when the concentration reached a certain level and the fixed amount would not increase.

It can also be seen from Figure 6 and Figure 7 that ssDNA junctions in the scribed region were not very uniform and orderly. There is still a gap in the goal of realizing the controllable, orderly and uniform functionalization of self-assembly films on silicon surfaces. The reasons for this were mainly due to the following aspects: (1) The self-assembled film generated by the diamond cutting tool in the first step was not very uniform and orderly; (2) The overall experimental efficiency was low due to the multi-step experimental operation; (3) The fixation effect of ssDNA was closely related to the pH value of the fixation solution, ionic strength, length and concentration of fixation probe, fixation time and other factors. Subsequent experiments in DNA probe fixation are needed to conduct precise research and analysis on the above problems and related factors.

## 4. Conclusions

In this work, we proposed a feasible strategy to immobilize ssDNA on silicon substrate using a mechano–chemical method. Firstly, diazo benzoate film was prepared, and AFM, X-ray photoelectron spectroscopy and infrared spectroscopy were used to detect and analyze the self-assembled film on a silicon surface. On the one hand, the existence of benzoate diazo salt self-assembled film was proven, and on the other hand, the aromatic hydrocarbon molecules in solution were covalently connected to the silicon substrate by Si–C bonds.

The principle of ssDNA immobilized on a silicon surface by a self-assembling membrane of aromatic diazonium salt was studied. Further experiments were carried out to connect single-stranded DNA to the silicon substrate. The XPS spectra of samples before and after fixing DNA showed that obvious N1s and P2p peaks appear after DNA assembly. The peak around 398.4 eV was the characteristic peak of N1s which were introduced because of the DNA modified with –NH_2_. The P2p peak around 130.6 eV was caused by the single stranded DNA base sequence. Meanwhile, the contents of C1s, O1s and Si2p were significantly changed after assembly. The content changes of main elements suggest that DNA has been attached to the silicon surface via the aromatic hydrocarbon coupling layer.

The fluorescence microscope was used to detect the fixed DNA in the self-assembled region. The results showed that ssDNA was successfully connected to the self-assembled film on the silicon surface. In order to prove that only functional areas on the silicon surface were connected with ssDNA, the silicon surface after functionalization was cleaned. The area with fluorescence signal was the part scribed in the solution, and the black areas on both sides were the unpainted parts. This was a good comparison to show that we can actually fix DNA on a Si substrate by the mechano–chemical method.

The influence of the concentration of ssDNA on the fixation effect was studied. Due to the spatial arrangement structure of DNA and the electrostatic repulsion between DNA molecules, the fixed amount of DNA increased with the increase in DNA concentration, the hybridization amount also increased, and the hybridization time was shortened. However, when the DNA reached a certain concentration, the hybridization amount decreased. At the same time, it was found that ssDNA junctions on the silicon surface were not very uniform, which was related to many factors, such as the inhomogeneity of the self-assembled coupling layer, the multi-step experimental operation and the pH value of the fixation solution.

The above experimental results show that it is feasible to connect ssDNA to a silicon substrate with aromatic diazo salt molecules as the medium. The technology uses a mechano–chemical method to process three-dimensional controllable structure on the silicon surface and self-assemble an aromatic hydrocarbon coupling layer at the same time. Single-stranded DNA can be connected to the 3D structure on the silicon surface through the coupling layer in a covalent binding mode. While fixing ssDNA, the size, position and shape can be highly controlled.

At present, this method is in the preliminary trial stage. In the future, further quantitative and applied experiments are needed to improve it. It is possible that the technology has important scientific value, theoretical significance and engineering application prospects for promoting the development of nano-devices, biosensors, disease diagnosis, environmental monitoring and other fields. Our team is trying to use this technology to detect microorganisms corroding on flood gates. It will have a wider application space in the future.

## Figures and Tables

**Figure 1 micromachines-14-01134-f001:**
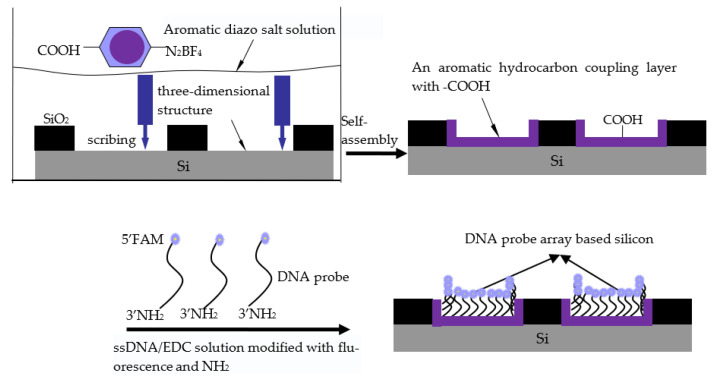
The principle of connecting ssDNA to silicon based on aryl diazonium salts.

**Figure 2 micromachines-14-01134-f002:**
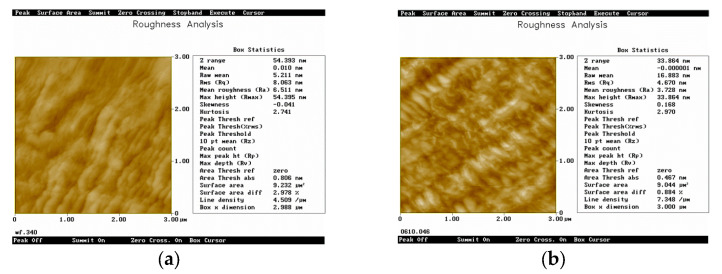
The roughness image before and after self-assembly in contact mode. (**a**) before self-assembly. (**b**) after self-assembly.

**Figure 3 micromachines-14-01134-f003:**
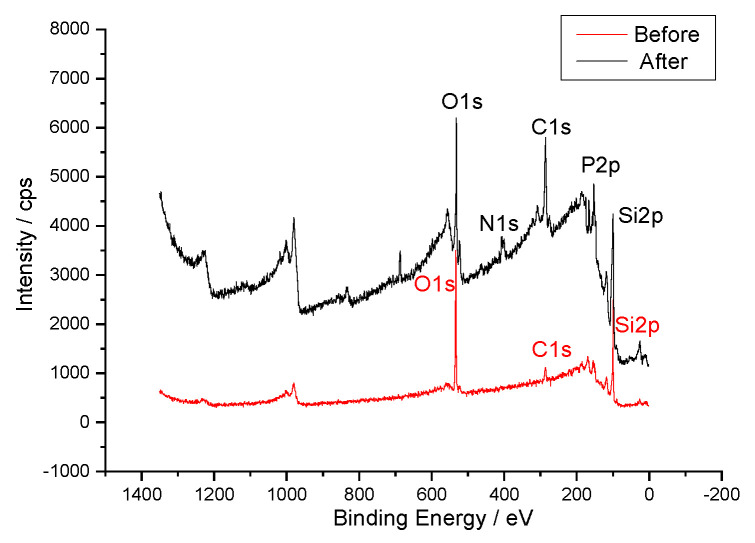
XPS spectra for sample before and after functionalizing DNA to the Si substrates.

**Figure 4 micromachines-14-01134-f004:**
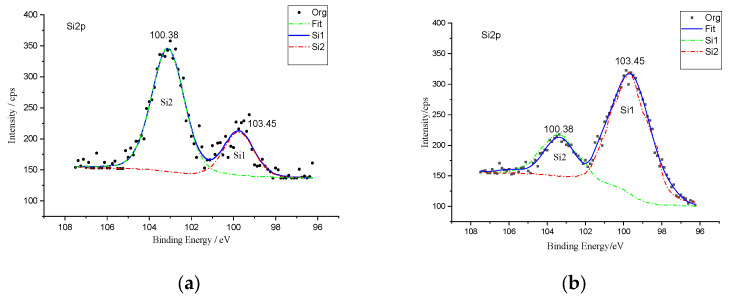
XPS spectra of Si2p for sample before and after self-assembly. (**a**) before self-assembly (**b**) after self-assembly.

**Figure 5 micromachines-14-01134-f005:**
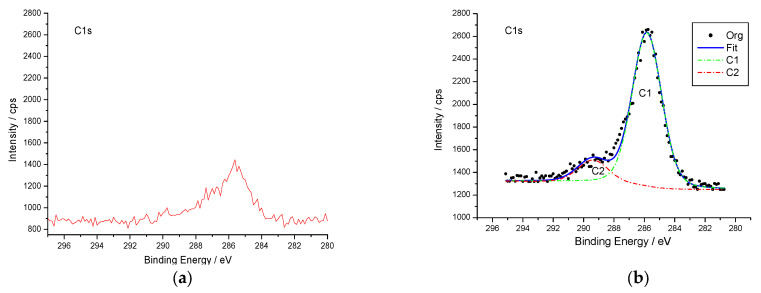
XPS spectra of C1s for sample before and after self-assembly. (**a**) before self-assembly. (**b**) after self-assembly.

**Figure 6 micromachines-14-01134-f006:**
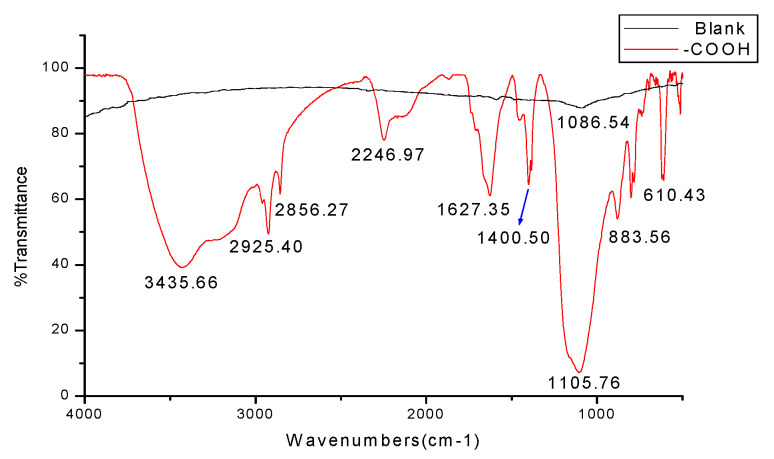
Using KBr detected the components of blank silicon and silicon surface modified with benzoic acid diazo salt.

**Figure 7 micromachines-14-01134-f007:**
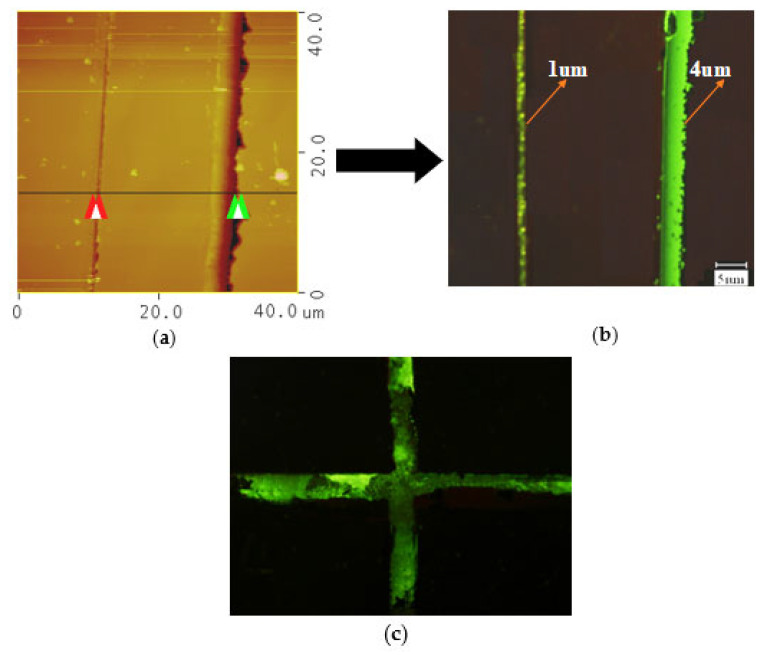
Schematic image of scribed regions connected with ssDNA. (**a**) AFM map before connection. (**b**) Fluorescent map after connection with DNA. (**c**) DNA three-dimensional cross structure.

**Figure 8 micromachines-14-01134-f008:**
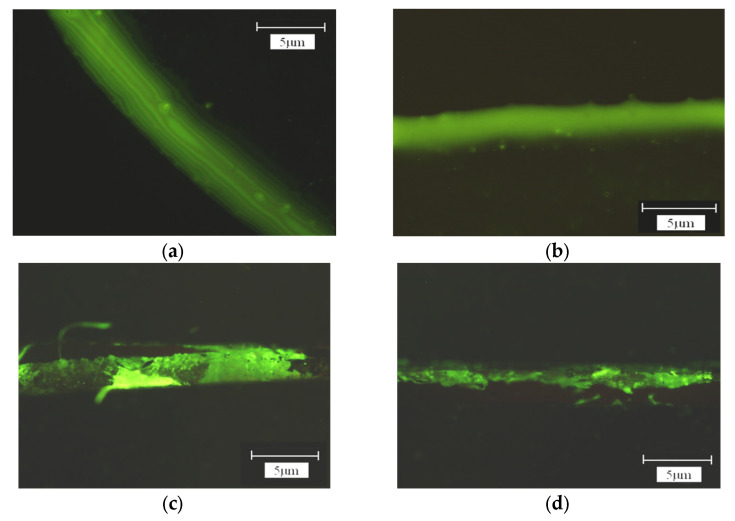
Schematic image of connected ssDNA at different concentrations. (**a**) 5 μmol/L. (**b**) 10 μmol/L. (**c**) 15 μmol/L (**d**) 20 μmol/L.

**Table 1 micromachines-14-01134-t001:** Element Content before and after functionalizing on Si Surface.

	Peak	Before	After
Peak ID	Center/eV	AT%	AT%
C1s	285.0	17.00	27.54
O1s	531.8	43.08	34.03
Si2p	100.1	39.92	30.52
N1s	398.4	0.00	3.25
P2p	130.6	0.00	4.66

## Data Availability

The data that support the findings of this study are available on request from the corresponding author.

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
