# Peer review of "Connection of ssDNA to Silicon Substrate Based on a Mechano–Chemical Method"

_micromachines, 2023, doi:10.3390/mi14061134_

Round 1

Reviewer 1 Report (New Reviewer)

1.      The introduction should be strengthened with the necessity of the application, material and method

2.      The introduction is short and should be expanded. Then highlight how your work is novel. Give specific and clear statements not general and ambiguous statement.

3.      The discussion section should be expanded to highlight the scientific contribution of this study to this field.

4.      To strengthen the discussion and justify the results, authors are suggested to go through some recent and very important reference papers and must include them in the revised manuscript.

  1. The authors should explain which exact problem could be solved by the present research.
  2. Moreover, the authors should formulate the present review work with respect to the other works in the field.

7.      The accuracy of the measurements of the technology should be presented.

8.      What are possible technology-oriented applications of the work for commercialization purposes?

9.      The authors should emphasize the contribution of work, future work, technology, or knowledge. The conclusion should be revised to be informative and show only significant findings of the study.

10.  Social implications shall be highlighted in the conclusion

11.  The English needed to be improved, and grammar needed to be checked carefully

Author Response

Thank you for your summary. We really appreciate your efforts in reviewing our manuscript. We have revised the manuscript accordingly. Our point-by-point responses are detailed below.

Point 1: The introduction should be strengthened with the necessity of the application, material and method.

Response 1: Thank you for your suggestion. As suggested by reviewer, the introduction section has been rewritten.

 Point 2: The introduction is short and should be expanded. Then highlight how your work is novel. Give specific and clear statements not general and ambiguous statement.

Response 2: We are grateful for the suggestion. As suggested by the reviewer, the introduction part has been expanded to clearly explain the features and novelty of the technology, and the relevant references in recent years are added and discussed.

 Point 3: The discussion section should be expanded to highlight the scientific contribution of this study to this field.

Response 3: We deeply appreciate the reviewer’s suggestion. According to your comment, in the discussion part, supplementary experiments were carried out. XPS full spectrum analysis (such as figure 3) and cleaning experiments (such as figure 7c)) after DNA fixation were added.

Point 4: To strengthen the discussion and justify the results, authors are suggested to go through some recent and very important reference papers and must include them in the revised manuscript.

Response 4: Thank you very much for your advice. In this paper, more important recent reference papers are added for reference and analysis, hoping to prove the rationality and feasibility of this technology.

Point 5: The authors should explain which exact problem could be solved by the present research.

Response 5: According to your comments, the introduction section has been rewritten to explain exactly the problem. Single-stranded DNA can be connected to the 3D structure on the silicon surface through the aromatic hydrocarbon coupling layer, and finally realize the connection of ssDNA probe. It can solve the problem that DNA probes are prepared in two-dimensional level at present.

Point 6: Moreover, the authors should formulate the present review work with respect to the other works in the field.

Response 6: The introduction part has been modified to supplement the research status of this field. The existing problems are analyzed, and the main research characteristics of this technology are summarized.

Point 7: The accuracy of the measurements of the technology should be presented.

Response 7: Since this paper is a preliminary experiment of DNA probe connection, the experimental results are all aimed at analyzing the feasibility of this technology, and a large number of repeated experiments are needed to verify the later application and accuracy of measurement. Thank you very much for your guidance on the future of this technology. We will make efforts to study in this direction.

Point 8: What are possible technology-oriented applications of the work for commercialization purposes?

Response 8: This technology can lay a foundation for the production and detection of three-dimensional DNA chips, and our team is studying the application of this technology to the detection of microbial corrosion of flood gates. In the future, it can also be applied to biosensors, disease diagnosis, environmental monitoring and other aspects.

 Point 9: The authors should emphasize the contribution of work, future work, technology, or knowledge. The conclusion should be revised to be informative and show only significant findings of the study.

Response 9: According to your comments, the conclusion has been revised to explain the research achievements and findings of this technology, as well as the contribution to future research in this field.

Point 10: Social implications shall be highlighted in the conclusion.

Response 10: According to your suggestions, the conclusion has been added to explain the scientific significance and social value of the technology.

Point 11: The English needed to be improved, and grammar needed to be checked carefully.

Response 11: Thank you very much for your careful review. According to your suggestions, part of the grammar in the paper has been checked and modified.

Reviewer 2 Report (New Reviewer)

Shi et al., presented a mechano/chemical approach to silicon surface modification technique for ssDNA attachment. This approach is interesting, and the characterization of the process was adequately provided. Overall, this paper is fascinating, but several improvements can be made to increase the overall quality and interest of the reader.

1. They should offer some comparison narrative of their proposed technique versus other present techniques, where the authors should demonstrate, with evidence, why their technique is better compared to existing Si surface modification techniques.

2. The word 'monolayer' is often thrown into papers reporting surface modification techniques, but very few could provide evidence that supports this claim. Therefore, I highly recommend that the author be cautious when using the term 'monolayer' in this work. Indeed diazonium can form a monolayer, but the author must show experimental evidence to support that.

3. Layer stability is critical in surface modification agents. The author did not provide any evidence to demonstrate their layer's stability.

4. Unify the figure's appearance; this is especially prominent in Figure 4.

5. Shi et al. performed a series of fluorescence microscopy post-ssDNA modification, but these results are all qualitative. I wish the author had provided some quantitative analysis of their image/modification of ssDNA on the surface.

6. Finally, the author should consider presenting some application of their technique or at least discuss it.

Author Response

Thank you for your summary. We really appreciate your efforts in reviewing our manuscript. We have revised the manuscript accordingly. 

Point 1: They should offer some comparison narrative of their proposed technique versus other present techniques, where the authors should demonstrate, with evidence, why their technique is better compared to existing Si surface modification techniques.
Response 1: We are grateful for the suggestion. As suggested by the reviewer, the introduction part has been rewrited to clearly explain the features and novelty of the technology, and the relevant references in recent years are added and discussed.

Point 2: The word 'monolayer' is often thrown into papers reporting surface modification techniques, but very few could provide evidence that supports this claim. Therefore, I highly recommend that the author be cautious when using the term 'monolayer' in this work. Indeed diazonium can form a monolayer, but the author must show experimental evidence to support that.
Response 2: I can’t agree more with your suggestion. As you said, the single layer can't be instrumented right now. Therefore, the ‘monolayer’ in the paper has been modified to film.

Point 3: Layer stability is critical in surface modification agents. The author did not provide any evidence to demonstrate their layer's stability.
Response 3: Thank you very much for your advice. Our team has been committed to researching mechanical and chemical modification of silicon surface for many years, and has also conducted relevant studies on the stability of self-assembled film and the influencing factors of film formation in the early stage. This paper mainly aims to verify the feasibility of DNA connection, and the experimental results discussed are all aimed at proving this goal. According to your suggestions, relevant research results will be published in the future in terms of stability and factors affecting the quality of film formation.

Point 4: Unify the figure's appearance; this is especially prominent in Figure 4.
Response 4: I’m very sorry for this problem. Thank you very much for your advice. According to your suggestion, the problem has been modified accordingly.

Point 5: Shi et al. performed a series of fluorescence microscopy post-ssDNA modification, but these results are all qualitative. I wish the author had provided some quantitative analysis of their image/modification of ssDNA on the surface.
Response 5: Thank you very much for your comments. According to your suggestion, we have carried out supplementary XPS experiments, and the experimental results are shown in Figure 3 and table 1, with corresponding elaboration. Since this paper is a preliminary experiment of DNA probe connection, the experimental results are all aimed at analyzing the feasibility of this technology, and a large number of repeated experiments are needed to verify the later application and accuracy of measurement. Thank you very much for your guidance on the future of this technology. We will make efforts to study in this direction.

Point 6: Finally, the author should consider presenting some application of their technique or at least discuss it.
Response 6: Thank you very much for your advice. This issue is supplemented in the introduction and conclusions. This technology can lay a foundation for the production and detection of three-dimensional DNA chips, and our team is studying the application of this technology to the detection of microbial corrosion of flood gates. In the future, it can also be applied to biosensors, disease diagnosis, environmental monitoring and other aspects.

Reviewer 3 Report (New Reviewer)

The present study on DNA functionalization on the Si substrate is incomplete and vague. Major corrections and proper justifications should be made against few points mentioned below. The present work does not qualify for publication.

1. Why the authors opt for directly functionalizing Si substrate instead of creating a thin film of gold which is well studied for functionalizing amino tagged ssDNA strands. Does it reduce the cost and complexity of the fabrication?

2. In the fabrication step it was not mentioned whether the unfunctionalized ssDNA strands were washed out or not and MCH could be used as the blocking agent.

3.      AFM topography of DNA functionalized Si substrate is essential in this work. It is interesting to find out whether the DNA agglomerates on the benzoate diazo salt functionalized silicon substrate.

4.      The peak shifts for C1s described in Figure 4 is confusing, major correction and proper clarification is required. The author should have kept the scale of X axis same for both graphs in Figure 4.

5.      In Figure 4(b) there should be another de-convoluted peak at around 287eV responsible for C-O. In line no 193, what does it mean “various impurity carbons adsorbed and mixed on the silicon surface”?

6.      XP spectra after functionalizing DNA molecules to the Si substrates should be include and further peak shifts for C1s, O1s, Si2p should be clearly mentioned with proper justification. There should be a notable change for these peaks.

Author Response

Thank you for your precious comments and advice. Those comments are all valuable and very helpful for revising and improving our paper, as well as the important guiding significance to our researches. We have studied comments carefully and have made correction which we hope meet with approval. Revised portion are marked in the paper. The main corrections in the paper and the responds to the reviewer’s comments are as flowing:

Point 1: Why the authors opt for directly functionalizing Si substrate instead of creating a thin film of gold which is well studied for functionalizing amino tagged ssDNA strands. Does it reduce the cost and complexity of the fabrication?

Response 1: We are grateful for the suggestion. The silicon substrate connects DNA through a thin film of gold, which is DNA fixation on a two-dimensional level. The characteristic of this technology is that silicon surface can be formed and functionalized simultaneously, and the shape and position of the DNA pattern produced can be controlled according to needs. If the technology is mature, it will be possible to fix DNA efficiently and cheaply.

 Point 2: In the fabrication step it was not mentioned whether the unfunctionalized ssDNA strands were washed out or not and MCH could be used as the blocking agent.

Response 2: According to your comments, the cleaning experiment was added to the last part of the fabrication step and the corresponding analysis was added to the discussion part. The results are shown in Figure 7c).

Point 3: AFM topography of DNA functionalized Si substrate is essential in this work. It is interesting to find out whether the DNA agglomerates on the benzoate diazo salt functionalized silicon substrate. 

Response 3: Thank you very much for your advice. Your suggestion is very good. We have also carried out AFM scanning on the samples after fixed DNA, hoping to make a more intuitive characterization. Whether it is because of the lack of DNA purity or the AFM resolution of our laboratory, the obtained scanning map is not very good and cannot clearly show the DNA morphology. Therefore, we carried out additional experiments with XPS and fluorescence microscopy. In the future, we will continue to improve the experimental conditions and strive to obtain clear DNA topography with AFM.

Point 4: The peak shifts for C1s described in Figure 4 is confusing, major correction and proper clarification is required. The author should have kept the scale of X axis same for both graphs in Figure 4.

Response 4: I am very sorry for the confusion caused to you. After supplementary experiments and data re-analysis, Figure 4 in the original paper is adjusted to Figure 5. And the Figure 5 has been corrected and elaborated.

Point 5: In Figure 4(b) there should be another de-convoluted peak at around 287eV responsible for C-O. In line no 193, what does it mean “various impurity carbons adsorbed and mixed on the silicon surface”? 

Response 5: Thank you very much for such meticulous and professional comments. According to your suggestion, we have conducted several experimental analyses, and the results are shown in Figure 5a) and b). It can been found that the peak around 285eV is caused by the impurity carbon inside the silicon wafer and the adsorption of atmospheric carbon elements, the peak around 287.8eV is the characteristic peak of carbon atom in C=O, and the peak around 288.9eV is the characteristic peak of carbon atom in -COOH. There is a problem in the supplementary experimental data. There is a small characteristic peak at 287.8ev, but the fitting result is not ideal. Then we will repeat the experiment to get the ideal characteristic peak.

 Point 6: XP spectra after functionalizing DNA molecules to the Si substrates should be include and further peak shifts for C1s, O1s, Si2p should be clearly mentioned with proper justification. There should be a notable change for these peaks. 

Response 6: According to your suggestion, we have carried out supplementary experiments, and the experimental results are shown in Figure 3 and table 1, with corresponding elaboration.

Round 2

Reviewer 1 Report (New Reviewer)

The revised version of the manuscript is good enough to be published. The authors addressed each comment in detail and made appropriate changes.

Reviewer 2 Report (New Reviewer)

The authors have answered all my questions.

Reviewer 3 Report (New Reviewer)

The manuscript is corrected as suggested and now can be accepted for publication in its present form.

This manuscript is a resubmission of an earlier submission. The following is a list of the peer review reports and author responses from that submission.

Round 1

Reviewer 1 Report

The present manuscript deals with the study of Benzoic Acid Diazo-Mediated Mechano-Chemical Method for 2 Immobilization of DNA Probes on Silicon Substrate. The manuscript is interesting and well presented. In my opinion, it should be revised.

1) Some of the experimental results should be included in the abstract.

2) In the introduction section, recent works should be discussed and cited.

3) Novelty of the work is not clear.

4) Figure 2, 3, 4 is not visible clearly.

5) What is 3.2. XPS detection & 3.3. FT-IR detection, 3.4. Fluorescence microscope detection ??

6) Conclusions should be rewritten.

7) There are some English and typo errors and it should be rectified.

Author Response

请参阅附件。

Reviewer 2 Report

This paper describes reaction of aryl diazonium salt and silicon substrate, which is followed by coupling with DNA. Their characterization is also described. Diazonium reaction is previously described in reference 24 by the same author, and DNA coupling is newly conducted here. Such explanation of this work is missing. It should be described what are novel aspects of this work. Without such discussions and evaluation of results, this manuscript is premature, and this referee cannot recommend publication in Micromachines.

Author Response

请参阅附件。

Reviewer 3 Report

The title does not refer to the content of the article at all. The authors should change it to something simpler. The authors do not write anything about immobilization, which is key in this matter. After all, it is known that the microarray technique basically consists of the following four steps: preparation of molecular probes and their immobilization on the surface of the plate, preparation of labeled DNA samples for analysis, hybridization of DNA samples with microarray probes, and finally reading the signals obtained from hybridization and data analysis. The authors did not present this in the paper. The results are not discussed with the scientific literature. In my opinion, the article will not find interest among readers, so I think it should be rejected. 

Round 2

Reviewer 2 Report

Revised manuscript provides a clearer view of this work. However, this study is premature, and property of the device has not been examined yet. For what purpose authors developed this device? Why the line shape is needed? What happens by the hybridization with the complimentary DNA? What is achieved using this device? This referee cannot recommend publication of this work in Micromachines.

Reviewer 3 Report

The article in the presented form is not suitable for publication. The manuscript is of very poor scientific quality with no substantiated data. Moreover, the authors dealt very superficially with the subject of presenting their results together with the current scientific literature. It looks very weak. In addition, the presented photos and charts are of poor quality, nothing comes of them. It is worth noting that the presented research does not bring anything new that could interest the readers.The methods shown are very old. The authors do not present anything new, but only use photos. It doesn't make any sense. The article cannot be accepted!